# Carbon Monoxide-Loaded Red Blood Cell Prevents the Onset of Cisplatin-Induced Acute Kidney Injury

**DOI:** 10.3390/antiox12091705

**Published:** 2023-09-01

**Authors:** Taisei Nagasaki, Hitoshi Maeda, Hiroki Yanagisawa, Kento Nishida, Kazuki Kobayashi, Naoki Wada, Isamu Noguchi, Ryotaro Iwakiri, Kazuaki Taguchi, Hiromi Sakai, Junji Saruwatari, Hiroshi Watanabe, Masaki Otagiri, Toru Maruyama

**Affiliations:** 1Department of Biopharmaceutics, Graduate School of Pharmaceutical Sciences, Kumamoto University, 5-1 Oe-Honmachi, Chuo-ku, Kumamoto 862-0973, Japan; t.nagasaki0507@gmail.com (T.N.); yanagisawa.hiroki@kao.com (H.Y.); 214y3001@st.kumamoto-u.ac.jp (K.N.); 217y2002@st.kumamoto-u.ac.jp (K.K.); 222y2006@st.kumamoto-u.ac.jp (N.W.); 220y3005@st.kumamoto-u.ac.jp (I.N.); 223y1008@st.kumamoto-u.ac.jp (R.I.); hnabe@kumamoto-u.ac.jp (H.W.); 2Division of Pharmacodynamics, Faculty of Pharmacy, Keio University, 1-5-30 Shibakoen, Minato-ku, Tokyo 105-8512, Japan; taguchi-kz@pha.keio.ac.jp; 3Department of Chemistry, Nara Medical University, 840 Shijo-cho, Kashihara 634-8521, Japan; hirosakai@naramed-u.ac.jp; 4Division of Pharmacology and Therapeutics, Graduate School of Pharmaceutical Sciences, Kumamoto University, 5-1 Oe-Honmachi, Chuo-ku, Kumamoto 862-0973, Japan; junsaru@gpo.kumamoto-u.ac.jp; 5Faculty of Pharmaceutical Sciences, Sojo University, 4-22-1 Ikeda, Nishi-ku, Kumamoto 860-0082, Japan; 6DDS Research Institute, Sojo University, 4-22-1 Ikeda, Nishi-ku, Kumamoto 860-0082, Japan

**Keywords:** carbon monoxide, red blood cell, acute kidney injury, cisplatin, oxidative stress, inflammation

## Abstract

Cisplatin-induced acute kidney injury (AKI) is an important factor that limits the clinical use of this drug for the treatment of malignancies. Oxidative stress and inflammation are considered to be the main causes of not only cisplatin-induced death of cancer cells but also cisplatin-induced AKI. Therefore, developing agents that exert antioxidant and anti-inflammatory effects without weakening the anti-tumor effects of cisplatin is highly desirable. Carbon monoxide (CO) has recently attracted interest due to its antioxidant, anti-inflammatory, and anti-tumor properties. Herein, we report that CO-loaded red blood cell (CO-RBC) exerts renoprotective effects on cisplatin-induced AKI. Cisplatin treatment was found to reduce cell viability in proximal tubular cells via oxidative stress and inflammation. Cisplatin-induced cytotoxicity, however, was suppressed by the CO-RBC treatment. The intraperitoneal administration of cisplatin caused an elevation in the blood urea nitrogen and serum creatinine levels. The administration of CO-RBC significantly suppressed these elevations. Furthermore, the administration of CO-RBC also reduced the deterioration of renal histology and tubular cell injury through its antioxidant and anti-inflammatory effects in cisplatin-induced AKI mice. Thus, our data suggest that CO-RBC has the potential to substantially prevent the onset of cisplatin-induced AKI, which, in turn, may improve the usefulness of cisplatin-based chemotherapy.

## 1. Introduction

Cisplatin is one of the most effective chemotherapeutic agents for the treatment of a broad spectrum of malignancies [1,2]. After being localized in the nucleus and binding to DNA, cisplatin gives rise to the formation of intrastrand DNA adducts and triggers the subsequent apoptosis of cancer cells [3,4,5]. However, cisplatin causes approximately 40 specific side effects [6], among which acute kidney injury (AKI) is a common clinical problem [7,8,9]. Patients with AKI generally have a high mortality rate in the range of 20–60% [10]. In addition, AKI is regarded as a sufficiently significant problem that limits the clinical use of cisplatin. Epidemiological studies have shown that about 33% of cancer patients who have been treated with cisplatin developed AKI [9]. In the case of patients with cisplatin-induced AKI, chemotherapy is frequently interrupted or changed to less effective chemotherapies that do not involve the use of cisplatin [9]. Currently, preventive measures using hydration and diuretics against cisplatin-induced AKI are employed in clinical settings to avoid the excessive exposure of cisplatin in the kidney [11]. However, since these measures also cause proximal renal tubule vacuolization via intense afferent arteriolar constriction [12], the prevalence of cisplatin-induced AKI is still high [13]. Therefore, developing a novel method that would prevent the onset of cisplatin-induced AKI is highly desirable.

After cisplatin is taken up by proximal tubular cells mainly via the organic cation transporter 2 (OCT2), it stimulates the production of reactive oxygen species (ROS) by increasing the expression of NADPH oxidase and inhibiting the mitochondrial electron transport chain [14]. The resulting ROS causes an increase in the production of inflammatory cytokines such as tumor necrosis factor-α (TNF-α) and interleukin-6 (IL-6) by activating the p38 mitogen-activated protein kinase pathway, leading to tubular cell apoptosis [15]. To address this issue, numerous researchers have investigated the usefulness of antioxidant or anti-inflammatory agents against cisplatin-induced AKI [16,17,18,19]. However, these agents also weaken the anti-tumor effects of cisplatin [20] since oxidative stress and inflammation contribute to the cisplatin-induced death of cancer cells [5,21]. Therefore, developing a therapeutic agent that exerts antioxidant and anti-inflammatory effects without weakening the anti-tumor effects of cisplatin is necessary.

Carbon monoxide (CO) is often referred to as a silent killer due to its high affinity for hemoglobin (Hb), resulting in CO poisoning. However, a number of studies have also reported that CO is an endogenous signaling molecule produced by heme oxygenase-mediated heme degradation [22]. Furthermore, it has also been reported that low concentration of CO has antioxidant [23,24] and anti-inflammatory effects [25,26]. Yoon, Y. et al. reported that CO-releasing molecule-3 (CORM-3) protects kidney epithelial cells from cisplatin-induced cytotoxicity [27]. Interestingly, it was also reported that CO enhances the anti-tumor effects of chemotherapeutic agents on various cancer cell types [28,29,30,31]. In support of this conclusion, Kawahara, B. et al. reported that photoactive CORM enhances the sensitivity of ovarian cancer cells to cisplatin by inhibiting the activity of cystathionine β-synthase, an enzyme that is involved in the survival of cancer cells [31]. Thus, the administration of exogenous CO is predicted to be a promising strategy for the prevention of cisplatin-induced AKI as well as for enhancing the anti-tumor effects of cisplatin.

To avoid CO poisoning, there is a need to develop a method that can safely supply CO to target organs [32]. The red blood cell (RBC) has a high biocompatibility and long half-time life [33]. It is also noteworthy that RBC has been reported to function as a dual carrier for CO as well as O_2_ in the blood [34]. Based on these facts, we developed a CO-loaded RBC (CO-RBC) in which CO is bound to nearly all of the Hb molecules in RBC [35,36,37,38,39]. Our previous studies reported on the renoprotective effects of CO-RBC on rodents in which AKI was induced by rhabdomyolysis [38] and ischemia-reperfusion (IR) [39]. These findings prompted us to hypothesize that CO-RBC might also exert renoprotective effects in cisplatin-induced AKI.

The aim of this study was to clarify the issue of whether CO-RBC attenuates the pathological conditions of cisplatin-induced AKI. We first investigated the protective effects of CO-RBC on cisplatin-induced cytotoxicity in human renal proximal tubule epithelial cells (HK-2 cells). We next prepared a mouse model of cisplatin-induced AKI and examined the therapeutic benefits of CO-RBC.

## 2. Materials and Methods

### 2.1. Cell Culture and Treatment

HK-2 cells and B16-F10 melanoma cells were purchased from the American Tissue Type Culture Collection (Manassas, VA, USA). HK-2 cells were grown in DMEM/F12 (Wako Pure Chemical, Ltd., Osaka, Japan) supplemented with 10% fetal bovine serum (Invitrogen, Carlsbad, CA, USA), penicillin (100 U/mL, Invitrogen, Waltham, MA, USA), and streptomycin (100 μg/mL, Invitrogen). Culture conditions were maintained at 37 °C in a humidified atmosphere at 5% CO_2_. HK-2 cells were treated with O_2_-RBC (1.0 × 10^7^ cell/mL) or CO-RBC (1.0 × 10^7^ cell/mL) for 3 h, followed by incubation with 25 μM cisplatin (Nippon Kayaku Co., Ltd., Tokyo, Japan) for 3 or 24 h. B16-F10 melanoma cells were cultured in DMEM supplemented with 10% fetal bovine serum, 100 U/mL penicillin, and 100 µg/mL streptomycin at 37 °C in 5% CO_2_. B16-F10 melanoma cells were treated with 100 μM cisplatin and O_2_-RBC (1.0 × 10^7^ cells/mL) or CO-RBC (1.0 × 10^7^ cells/mL) for 24 h.

### 2.2. WST-8 Assay

The cell viability was assessed using a Cell Counting Kit-8, including WST-8 (Dojindo Lab, Kumamoto, Japan). HK-2 cells were seeded in 96-well plates. The cells were treated with O_2_-RBC (1.0 × 10^7^ cell/mL) or CO-RBC (1.0 × 10^7^ cell/mL) for 3 h, then incubated with 25 μM cisplatin for 24 h. WST-8 reagent was added to 96-well plates, and the plates were maintained at 37 °C for 2 h. The absorbance of each well was determined at 450 nm with a microplate reader (Bio-Rad, Hercules, CA). Cell morphological changes were observed under a light microscope using a microscope system (Nikon Eclipse TS 100, Nikon, Tokyo, Japan).

### 2.3. TUNEL Staining

To evaluate apoptosis, HK-2 cells and kidney sections were stained with TUNEL reagent using an in situ cell death detection kit (Roche, Basel, Switzerland). HK-2 cells were grown on a glass-based dish (IWAKI, Shizuoka, Japan). The cells were treated with O_2_-RBC (1.0 × 10^7^ cell/mL) or CO-RBC (1.0 × 10^7^ cell/mL) for 3 h, followed by incubation with 25 μM cisplatin for 3 h. The cells and kidney sections were also stained with DAPI (Dojin Chemical, Kumamoto, Japan). The image of sections was obtained using a BZ-X710 instrument (Keyence, Osaka, Japan).

### 2.4. Measurement of ROS Levels

Intracellular ROS levels in the HK-2 cells were determined using a CM-H_2_DCFDA probe (Thermo Fisher Scientific, Waltham, MA, USA). HK-2 cells were grown on a glass-base dish or seeded in 96-well plates. The cells were treated with O_2_-RBC (1.0 × 10^7^ cell/mL) or CO-RBC (1.0 × 10^7^ cell/mL) for 3 h, followed by incubation with 25 μM cisplatin for 3 h. After incubation of Dulbecco’s phosphate buffered saline containing 2 μM CM-H_2_DCFDA probe at 37 °C for 30 min, imaging of the cells was visualized using a BZ-X710 instrument. Each fluorescence intensity of cells was measured at excitation 490 nm and emission 540 nm by using a microplate reader (Synergy H1, Bio Tek Instruments Inc., Winooski, VT, USA).

### 2.5. Immunostaining Analysis

To conduct immunofluorescence staining with anti-myeloperoxidase (MPO, 1:50, Santa Cruz Biotechnology Inc., sc-16128-R, Santa Cruz, CA, USA), kidney sections were treated with HistoVT One (Nacalai Tesque, Kyoto, Japan) for antigen retrieval at 95 °C for 30 min, then blocked using 4% Block Ace (KAC, Kyoto, Japan) at 37 °C for 30 min. The kidney sections were reacted with the primary antibody at 4 °C overnight, then reacted with an Alexa Fluor 647 anti-rabbit IgG antibody (H + L) at room temperature for 1.5 h. To perform the immunohistochemical staining with anti-nitrotyrosine (NO-Tyr, 1:50, Merck Millipore, AB5411, Burlington, MA, USA), anti-4-hydroxynonenal (4-HNE, 1:50, Bioss Antibodies, bs-6313R, Woburn, MA, USA), or anti-F4/80 (1:50, Thermo Fisher Scientific, 14-4801-82), antigens were retrieved. Kidney sections were reacted with 3% H_2_O_2_/methanol solution at room temperature for 30 min to inhibit the action of endogenous peroxidase and were reacted with the primary antibody at 4 °C overnight. The kidney sections were treated with peroxidase-conjugated anti-rabbit IgG (Histofine Simple Stain MAX-PO, Nichirei Biosciences, Tokyo, Japan) at room temperature for 30 min, followed by reaction with diaminobenzidine solution at room temperature for 3 min. All images of slides were acquired with a BZ-X710 instrument. The positive area of 4-HNE and NO-Tyr or cell number of F4/80 and MPO were qualified in an area of 270 × 360 µm (original magnification, ×400). At minimum, ten areas of kidney specimens from each mouse were used in the analysis.

### 2.6. Quantitative Real-Time Polymerase Chain Reaction (qRT-PCR) Analysis

Isolation of total RNA from HK-2 cells or kidney tissue and the qRT-PCR were performed as previously described [39]. The gene expression for Kim-1, TNF-α, and IL-6 were measured by qRT-PCR. All primers were purchased from Takara Bio (Tokyo, Japan). The sequences of the oligonucleotide primers are provided in Appendix A.

### 2.7. Cisplatin-Induced AKI Mice Model

ICR mice (male, 4 weeks, 20–22 g, SLC Japan, Inc., Shizuoka, Japan) were used in all experiments, and the animals were housed in a room with food and drinking water on a 12-h light/dark cycle while maintaining the temperature (18–24 °C) and relative humidity (40–70%). All animal experiments were approved by the Animal Experiments of Kumamoto University and conducted in accordance with the National Institutes of Health. Before each experiment, all mice were housed in a conventional room to allow them to acclimatize for one week. Mice received an intravenous injection of saline, O_2_-RBC (1400 mgHb/kg), or CO-RBC (1400 mgHb/kg) 90 min before the intraperitoneal administration of cisplatin (15 mg/kg). The mice were then sacrificed at 96 h after the administration of cisplatin. Blood samples were collected to determine the values for blood urea nitrogen (BUN) and serum creatinine (SCr). The right kidney was analyzed for the qRT-PCR, and the left kidney was fixed in phosphate-buffered 4% formalin for histological examination.

### 2.8. Preparation of O_2_-RBC and CO-RBC

The solution of O_2_-RBC and CO-RBC (100 mgHb/mL) was purified as described in a previous report [35]. The carboxyhemoglobin (HbCO) level was measured as the percentage of Hb that is bound to CO and is commonly used as an indicator of CO exposure [40]. Based on a report by Park, J. et al. [41], the HbCO level in the CO-RBC solution was measured using a spectroscopic method based on absorptions at 419 (HbCO) and 415 nm (deoxyHb). In each experiment, the HbCO level in the CO-RBC solution was confirmed to be approximately 100% after 10 min of CO gas bubbling. The actual concentration of CO in the CO-RBC solution was measured by gas chromatography.

The CO-RBC dose of 1.0 × 10^7^ cells/mL in in vitro experiments was used based on a report by Kwon, S. et al. [42]. On the other hand, the CO-RBC dose of 1400 mgHb/kg in the in vivo experiments was the safest maximum dose of Hb in mice that can be administered [43,44]. A solution of 100 mgHb/mL, which contains 6.0 × 10^9^ RBC cells/mL, was prepared to administer the CO-RBC to mice. To adjust the dose to 1.0 × 10^7^ cells/mL of RBC, a solution of 6.0 × 10^9^ cells/mL of RBC was diluted 600-fold. Body weight is not the only factor that influences the scaling for dose calculations. The correction factor (K_m_) was estimated by dividing the average body weight (kg) of the species by its body surface area (m^2^) [45]. Considering the K_m_ of mice (3) and humans (37), the CO-RBC dose for humans was calculated to be 113 mgHb/kg (=1400 mgHb/kg × K_m_ (mice)/K_m_ (humans)). Assuming that an adult human weighing 60 kg receives a single injection of CO-RBC (113 mgHb/kg), 6.7 gHb containing 1.6 mg CO (0.02 mg/kg) would have been administered.

### 2.9. Analysis of Renal Function

To obtain plasma samples, blood from the mice was centrifuged at 6000 rpm at room temperature for 10 min, and the supernatant was then collected. The values for BUN and SCr in plasma were measured using a FUJI DRI-CHEM 7000 (Fujifilm, Tokyo, Japan).

### 2.10. Histologic Examination of Kidney Tissue

Kidney sections at 2 µm embedded in paraffin were prepared. Histologic examination was performed by staining of periodic acid-Schiff (PAS). Stained sections were observed with a BZ-X710 instrument.

### 2.11. Diacron-Reactive Oxygen Metabolites (d-ROMs) Test

The d-ROM levels in plasma were measured using an analyzer of free radicals (FREE carpe diem, Wismerll, Tokyo, Japan).

### 2.12. Quantification of CO Levels in Kidney and Brain

To completely dissociate CO from Hb or hemoprotein, the whole kidney or brain was incubated with saponin (5 mL) in a vial (10 mL) at room temperature for 4 h. The levels of CO in the headspace of the vial were quantified by gas chromatography with a TRIlyzer mBA-3000 (Taiyo Instruments, Inc., Osaka, Japan).

### 2.13. Statistics

The data are expressed as the mean ± SE. Two group data were compared using the Student’s *t*-test. More than two groups were compared using analysis one-way ANOVA followed by Tukey’s multiple comparisons.

## 3. Results

### 3.1. CO-RBC Alleviates Cisplatin-Induced Cytotoxicity in HK-2 Cells

To investigate the cytoprotective effects of CO-RBC on cisplatin-induced cell death, we pre-treated HK-2 cells with a vehicle, O_2_-RBC (1.0 × 10^7^ cells/mL), or CO-RBC (1.0 × 10^7^ cells/mL) for 3 h. After the pretreatment, the cells were treated with 25 μM cisplatin for 24 h. Bright-field images obtained from microscopic analyses showed that the decreased cell viability caused by cisplatin was restored in the case of the CO-RBC treatment (Figure 1A). On the contrary, the O_2_-RBC treatment had no effect on the cell viability in the cisplatin-treated cells. To further confirm this cytoprotective effect of CO-RBC, we measured cell viability using a WST-8 assay (Figure 1B). Since tubular cell apoptosis is a predominant feature of cisplatin-induced cytotoxicity [46], we performed TUNEL staining to evaluate the anti-apoptotic effects of CO-RBC on cisplatin-induced apoptosis in HK-2 cells. The cisplatin treatment increased the number of apoptosis cells, but such increases were suppressed by the CO-RBC treatment (Figure 1C). In addition, the CO-RBC treatment suppressed the cisplatin-induced levels of gene expression for kidney injury molecules (Kim-1), a marker for proximal tubular cell injury [47] (Figure 1D).

### 3.2. CO-RBC Exerts Antioxidant and Anti-Inflammatory Effects on Cisplatin-Treated HK-2 Cells

We also investigated the antioxidant effects of CO-RBC using CM-H_2_DCFDA for the detection of intracellular ROS. The CO-RBC treatment resulted in a substantial decrease in the amount of intracellular ROS that was induced by the cisplatin treatment. The O_2_-RBC treatment, however, failed to exert such an antioxidant effect (Figure 2A,B). We further examined the anti-inflammatory effects of CO-RBC. qPCR experiments showed that the CO-RBC treatment suppressed the cisplatin-induced levels of gene expression for TNF-α and IL-6 (Figure 2C,D).

### 3.3. CO-RBC Exerts a Renoprotective Effect on Cisplatin-Induced AKI Mice

To evaluate the therapeutic efficacy of CO-RBC in vivo, we prepared a mouse model of cisplatin-induced AKI according to a report by Kodama, A. et al. [48]. Based on our previous report in which CO levels in the kidney of normal mice were increased at 90 min after the administration of CO-RBC [39], a single dosage of CO-RBC (1400 mgHb/kg) was administrated 90 min prior to the cisplatin treatment (15 mg/kg) (Figure 3A). The intraperitoneal administration of cisplatin resulted in an elevation in the values for BUN and SCr in plasma, both of which are indicators of kidney function (Figure 3B,C). A prophylactic administration of CO-RBC significantly suppressed these elevations, while no improvement was detected in the case of the administration of O_2_-RBC (Figure 3B,C). These findings indicate that the CO that is liberated from CO-RBC exerts a renoprotective effect on cisplatin-induced AKI.

### 3.4. CO-RBC Protects the Kidney from Cisplatin-Induced Damage

Tubular cells are particularly vulnerable to cisplatin-induced cytotoxicity [49]. Thus, we further examined the issue of whether CO-RBC alleviates the cisplatin-induced injury to these types of cells. The levels of gene expression for Kim-1 that were induced by cisplatin were significantly suppressed in the case of a prophylactic administration of CO-RBC (Figure 4A). To evaluate histological features and renal apoptosis, we performed PAS staining and TUNEL staining. The cisplatin-treated mice had an increase in the number of columnar formation and apoptosis-positive cells, whereas a CO-RBC prophylactic administration improved such unfavorable features (Figure 4B).

### 3.5. CO-RBC Exerts Antioxidant Effects on Cisplatin-Induced AKI Mice

To examine the antioxidant effects of CO-RBC on cisplatin-induced AKI, we performed immunostaining of kidney tissue using antibodies against 4-HNE or NO-Tyr, both of which serve as oxidative stress markers [50,51]. Kidney tissue with disorders caused by cisplatin showed positive areas for 4-HNE and NO-Tyr, especially in the vicinity of tubules, whereas such positive areas were reduced by the prophylactic administration of CO-RBC (Figure 5A). The level of d-ROMs in plasma also serves as an oxidative stress biomarker for AKI [52]. Consistent with the immunostaining results for 4-HNE and NO-Tyr, a prophylactic administration of CO-RBC attenuated the cisplatin-induced levels of d-ROMs in plasma (Figure 5B).

### 3.6. CO-RBC Exerts Anti-Inflammatory Effects on Cisplatin-Induced AKI Mice

We further measured the levels of the gene expression for TNF-α and IL-6 in the kidney. The increased levels of these genes caused by cisplatin were suppressed in the prophylactic administration of CO-RBC (Figure 6A,B). Inflammatory cells such as macrophages and neutrophils infiltrate kidney tissue and, by secreting TNF-α and IL-6, play a significant role in the development of cisplatin-induced AKI [9,53]. Immunostaining using antibodies against F4/80 (a typical macrophage marker) or MPO (a typical neutrophil marker) indicated that the increased renal infiltration of macrophages and neutrophils caused by cisplatin was suppressed by the prophylactic administration of CO-RBC (Figure 6C).

### 3.7. CO-RBC Does Not Weaken the Anti-Tumor Effect of Cisplatin in B16-F10 Melanoma Cells

Therapeutic agents that exert antioxidant and anti-inflammatory effects without weakening the anti-tumor effects of cisplatin are helpful for patients who suffer from cisplatin-induced AKI. To determine whether CO-RBC affects the anti-tumor effects of cisplatin, B16-F10 melanoma cells were co-treated with 100 μM cisplatin and vehicle, O_2_-RBC (1.0 × 10^7^ cells/mL), or CO-RBC (1.0 × 10^7^ cells/mL) for 24 h. As a result, cisplatin-reduced cell viability was not affected by co-treatment with CO-RBC (Figure 7). This result suggests that CO-RBC does not weaken the anti-tumor effect of cisplatin and is a useful drug when used in combination with cisplatin. However, our experiment with B16-F10 melanoma cells is not sufficient to provide rigorous evidence. In the future, the effects of CO-RBC on the anti-tumor effect of cisplatin in tumor-bearing mice should be confirmed.

## 4. Discussion

In the present study, we examined the renoprotective effects of CO-RBC on cisplatin-induced renal injury both in vitro and in vivo. CO-RBC attenuated cisplatin-induced cytotoxicity in HK-2 cells. In addition, CO-RBC mitigated proximal tubular cell injury via the attenuation of oxidative stress and inflammation in a mouse model of cisplatin-induced AKI, resulting in the improvement of the deteriorated kidney function and renal histology. These data suggested that CO-RBC has the potential to substantially prevent the onset of cisplatin-induced AKI.

To realize the clinical application of CO, ruthenium (Ru)-based CO-releasing agents, CORMs, have been developed [54]. The advantage of these agents is that the CO dosage can be easily adjusted based on the amount of Ru being used. However, since CORMs have a short half-life, frequent administration of CORMs is required for achieving adequate therapeutic effects [32]. Actually, Tayem, Y. et al. showed that the daily administration of CORM-3 for 4 days improved the pathological condition of cisplatin-induced AKI [55]. In this study, we used CO-RBC, which has prolonged release properties of CO [39], and demonstrated that a single dose of CO-RBC exerted renoprotective effects on cisplatin-induced AKI (Figure 3, Figure 4, Figure 5 and Figure 6). This result is consistent with our previous study using CO-loaded Hb vesicles [56]. Thus, our therapeutic approach using an RBC-based CO-releasing agent is expected to be clinically applicable for the prevention of cisplatin-induced AKI.

CO is produced during the metabolism of heme by the action of heme oxygenase-1 (HO-1) [57]. Since heme proteins, such as those in RBC, upregulate the protein levels of HO-1, resulting in an enhanced heme metabolism and the subsequent production of CO [58], the possibility that heme proteins in RBC also exert renoprotective effects cannot be excluded. To determine whether the CO liberated from CO-RBC is responsible for the renoprotective effects on cisplatin-induced AKI, we carefully analyzed the kidneys obtained from cisplatin-induced AKI mice after the administration of saline (control), O_2_-RBC (1400 mgHb/kg), or CO-RBC (1400 mgHb/kg) (Appendix A). The administration of O_2_-RBC had no effect on cisplatin-elevated levels of gene expression for Kim-1, TNF-α, and IL-6 and the levels of d-ROMs (Appendix A). Consistent with these results, O_2_-RBC failed to improve cisplatin-induced histopathologic disorders that were analyzed by histological staining (PAS and TUNEL) and immunostaining (4-HNE, NO-Tyr, F4/80, and MPO) (Appendix A). These findings suggest that the 1400 mgHb/kg of RBC used in the experiments was an insufficient amount to exert renoprotective effects on cisplatin-induced AKI. We, therefore, concluded that the CO derived from CO-RBC exerted renoprotective effects on cisplatin-induced AKI.

The control of mitochondrial ATP production, which is required for tissue repair, is known as a representative metabolic effect of CO [59]. Lavitrano M. et al. reported that the hearts of CO-treated pigs that were subjected to cardiopulmonary bypass surgery with cardioplegic arrest showed significantly higher ATP levels, resulting in a reduction in the production of apoptotic cells [60]. Further, CO shifts energetic metabolism from glycolysis to oxidative phosphorylation-dependent ATP production in endothelial cells [61]. We recently showed that the administration of CO-RBC resulted in increased ATP levels via the activation of AMP-activated protein kinase in the kidney of IR-induced AKI or an AKI to CKD transition model mice [39]. Given that the pathogenesis of cisplatin-induced AKI is associated not only with oxidative stress and inflammation but also with impaired mitochondrial metabolism [46], there is the possibility that CO-mediated mitochondrial metabolism could be attributed to the renoprotective effects of CO-RBC on cisplatin-induced AKI.

Proximal tubular cell injury is elicited by oxidative stress and inflammation that are induced by cisplatin [14,15]. Yan, W. et al. reported that astragaloside IV, a traditional Chinese herb medicine, attenuated cisplatin-induced intracellular ROS levels and inflammatory cytokines in HK-2 cells and then improved tubular cell injury [62]. Consistent with this report, we showed that CO-RBC also exerted such beneficial effects not only in vitro but also in vivo (Figure 1, Figure 2, Figure 5, and Figure 6). These findings indicate that the renoprotective effects of CO-RBC on cisplatin-induced AKI are due to the amelioration of proximal tubular cell injury via its antioxidant and anti-inflammatory effects.

Since tubular injury is a key pathology associated with the nephrotoxic effects of cisplatin [63], the present study used HK-2 cells to investigate the renoprotective effect of CO-RBC on cisplatin-induced AKI. On the other hand, endothelial cells and macrophages are also involved in the pathogenesis of AKI [64]. CO has excellent cell membrane permeability, thus allowing it to easily gain access to cells regardless of the cell type being considered. Ruan M. et al. reported that a CORM-2 pretreatment significantly prevented IR–induced endothelial injury in IR-AKI mice [65]. Additionally, we also recently reported that CO shifts the polarity of macrophages toward an anti-inflammatory phenotype [66]. We, therefore, speculate that CO can be expected to comprehensively exert a renoprotective effect via acting not only on tubular cells, which are the origin of renal pathology induced by cisplatin but also on a variety of other cells, such as endothelial cells and macrophages.

Among the various OCT isoforms, OCT2 plays a crucial role in the uptake of cisplatin in the kidney and contributes to the onset of cisplatin-induced AKI [67]. Clinical studies have shown that the severity of cisplatin-induced AKI diminishes in patients with a mutation in the OCT2 gene [68,69,70]. Based on these findings, therapeutic drugs targeting OCT2 for cisplatin-induced AKI are currently being developed [71,72,73]. Kim H. et al. reported that glutamine attenuated cisplatin-induced AKI by inhibiting the expression of OCT2 [74]. In this study, we did not investigate the possibility that CO-RBC affects the expression of OCT2. In addition, there is no evidence of the existence of a relationship between CO and OCT2, as well as other isoforms of OCT. In the future, it will be necessary to address the issue of whether CO-RBC affects OCT2 expression and the amount of cisplatin in the kidney of cisplatin-treated mice.

As of this writing, clinical trials have been performed to prevent the onset of cisplatin-induced AKI by pre-administrating magnesium or mannitol (https://clinicaltrials.gov/) (accessed on 25 June 2015 or 5 February 2020). Despite these measures, it is difficult to prevent AKI in patients who receive cisplatin [63]. In preclinical studies, some researchers have investigated the usefulness of some drugs, e.g., stimulator of interferon genes antagonist [75], huaiqihuan extractum [76], and human serum albumin-thioredoxin fusion protein [48], against cisplatin-induced AKI. They designed a pre-administration schedule for these drugs, assuming a therapy that was combined with cisplatin-based chemotherapy. Based on these reports, we also prophylactically administrated CO-RBC to cisplatin-treated mice in our study (Figure 3A). These findings confirm that the present study reflects the clinical situation appropriately.

Oxidative stress is involved in the development of cisplatin-induced ototoxicity and neurotoxicity, which are also known side effects of cisplatin [77,78]. However, drug transport in the ear and brain is strictly obstructed by the blood-labyrinth barrier and the blood-brain barrier, respectively [79,80]. Conveniently, CO gas can freely pass through the plasma membrane and then diffuse to various organs [81]. In fact, we showed that the intravenous administration of CO-RBC to normal mice resulted in an increased level of CO in the brain as well as the kidney (Appendix A). Moreover, the antioxidant effects of CO-RBC have been well-defined in various animal models [36,37,38,39]. Therefore, CO derived from CO-RBC has the potential to be distributed to even the ear and brain, thus subsequently improving cisplatin-induced neurotoxicity and ototoxicity via its antioxidant effects.

Since favorable results from in vitro experiments are not always replicated in in vivo experiments [82], our experiment with B16-F10 melanoma cells is not sufficient to provide rigorous evidence that CO-RBC does not weaken the anti-tumor effect of cisplatin. It has been reported by other investigators that CO enhances the anti-tumor effects of chemotherapeutic agents [28,29,30,31]. Actually, CO suppresses angiogenesis and inhibits the activity of cystathionine β-synthase, which is important in regulating cancer cell homeostasis, leading to synergistic effects when combined with cisplatin or doxorubicin [83]. Therefore, it is possible that CO-RBC could also enhance the anti-tumor effects of cisplatin. In the future, the effects of CO-RBC on the anti-tumor effect of cisplatin in tumor-bearing mice should be confirmed.

Approximately 50% of cancer patients receive cisplatin-based chemotherapy [84], and such chemotherapy regimens require many procedures [85]. To reduce the burden on healthcare workers and cancer patients during cisplatin-based chemotherapy, it is desirable to develop a drug that can simultaneously provide both CO and cisplatin. In recent years, taking advantage of characteristics of RBC, such as the massive interior space [86], attempts have been made to use RBC as a carrier for the delivery of anticancer drugs [87,88,89]. Lucas A. et al. demonstrated that doxorubicin (Dox) -loaded RBC exerts a remarkable anti-tumor effect as compared with Dox alone and reduces the side effects of Dox such as cardiotoxicity and myelosuppression [90]. Given that Dox-loaded RBC was prepared without affecting the Hb content of RBC [91] or RBC deformability [92], CO- and cisplatin-loaded RBC may be a practical cell therapy with less cytotoxicity.

## 5. Conclusions

The findings reported in this study showed that a prophylactic administration of CO-RBC exerts significant renoprotective effects on cisplatin-induced AKI. This study points out the therapeutic potential of CO-RBC for the prevention of cisplatin-induced AKI, which, in turn, may improve the usefulness of cisplatin-based chemotherapy.

## Figures and Tables

**Figure 1 antioxidants-12-01705-f001:**
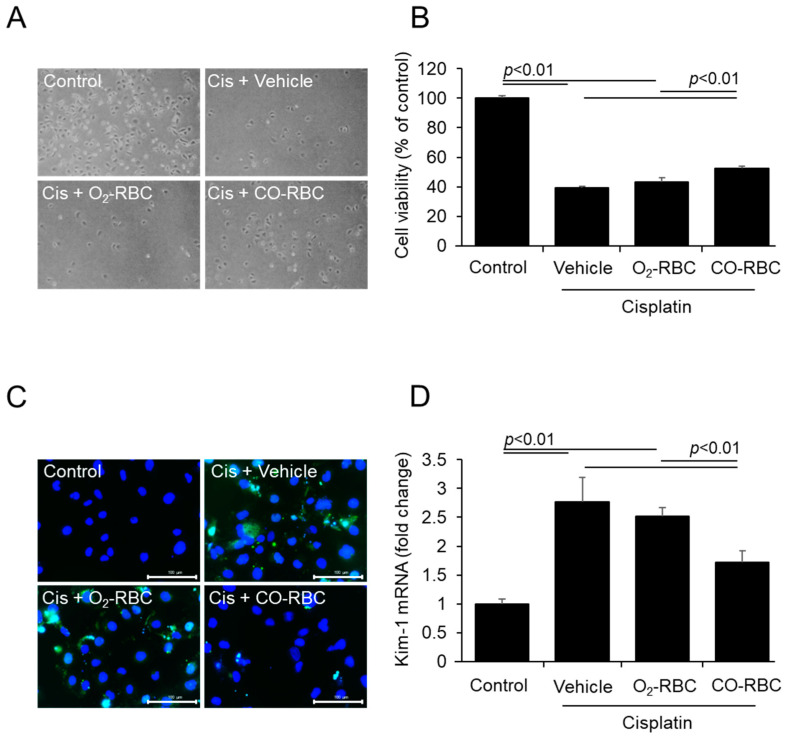
CO-RBC alleviates cisplatin-induced cytotoxicity in HK-2 cells. To investigate the cytoprotective effects of CO-RBC on cell viability, HK-2 cells were treated with vehicle, O_2_-RBC (1.0 × 10^7^ cell/mL), or CO-RBC (1.0 × 10^7^ cell/mL) for 3 h, followed by 25 μM cisplatin for 24 h. The cells were analyzed by (**A**) bright field microscope (magnification, ×40) and (**B**) WST-8 assay (*n* = 4/group). To assess the effect of CO-RBC on apoptosis and tubular cell injury, HK-2 cells were treated with vehicle, O_2_-RBC (1.0 × 10^7^ cell/mL), or CO-RBC (1.0 × 10^7^ cell/mL) for 3 h, followed by 25 μM cisplatin for 3 h. The cells were analyzed by (**C**) TUNEL staining (magnification, ×400, scale bars, 100 µm) and (**D**) qRT-PCR (*n* = 4/group). Results expressed as the mean ± S.E.M.

**Figure 2 antioxidants-12-01705-f002:**
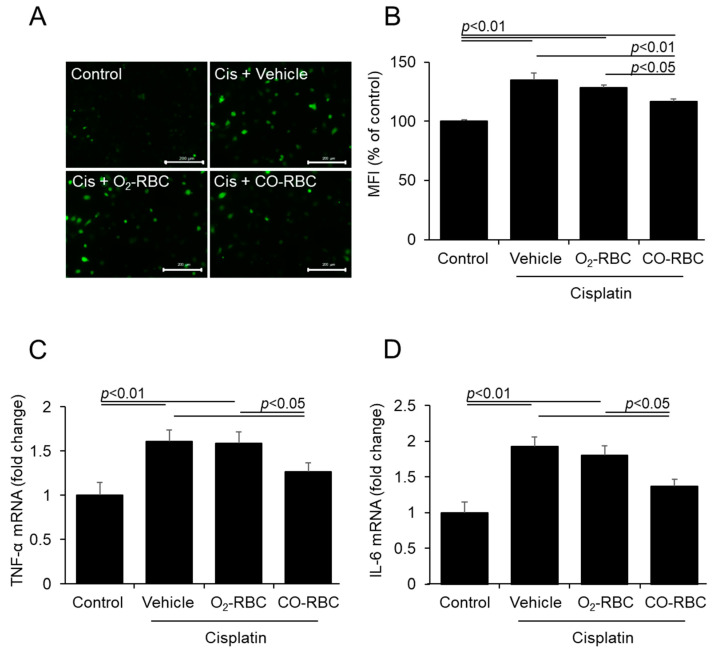
CO-RBC exerts antioxidant and anti-inflammatory effects on cisplatin-treated HK-2 cells. To assess the antioxidant and anti-inflammatory effects of CO-RBC, HK-2 cells were treated with vehicle, O_2_-RBC (1.0 × 10^7^ cell/mL), or CO-RBC (1.0 × 10^7^ cell/mL) for 3 h, followed by 25 μM cisplatin for 3 h. The cells were analyzed by (**A**,**B**) CM-H_2_DCFDA assay (magnification, ×200, scale bars, 200 µm, *n* = 4/group) and (**C**,**D**) qRT-PCR. (*n* = 4/group). Results expressed as the mean ± S.E.M.

**Figure 3 antioxidants-12-01705-f003:**
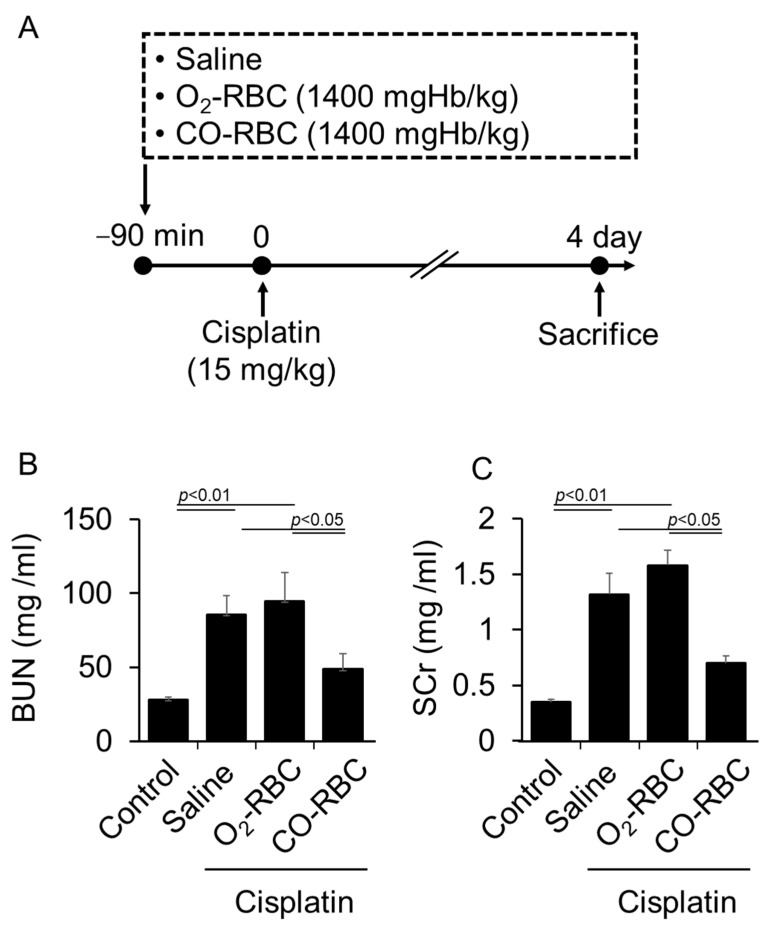
CO-RBC exerts a renoprotective effect on cisplatin-induced AKI. (**A**) A schematic summary of the administration schedule used for saline, O_2_-RBC (1400 mgHb/kg), or CO-RBC (1400 mgHb/kg) against cisplatin-induced AKI. (**B**,**C**) To measure kidney function, the values for BUN and SCr in plasma were assessed (*n* = 5/group). Results expressed as the mean ± S.E.M.

**Figure 4 antioxidants-12-01705-f004:**
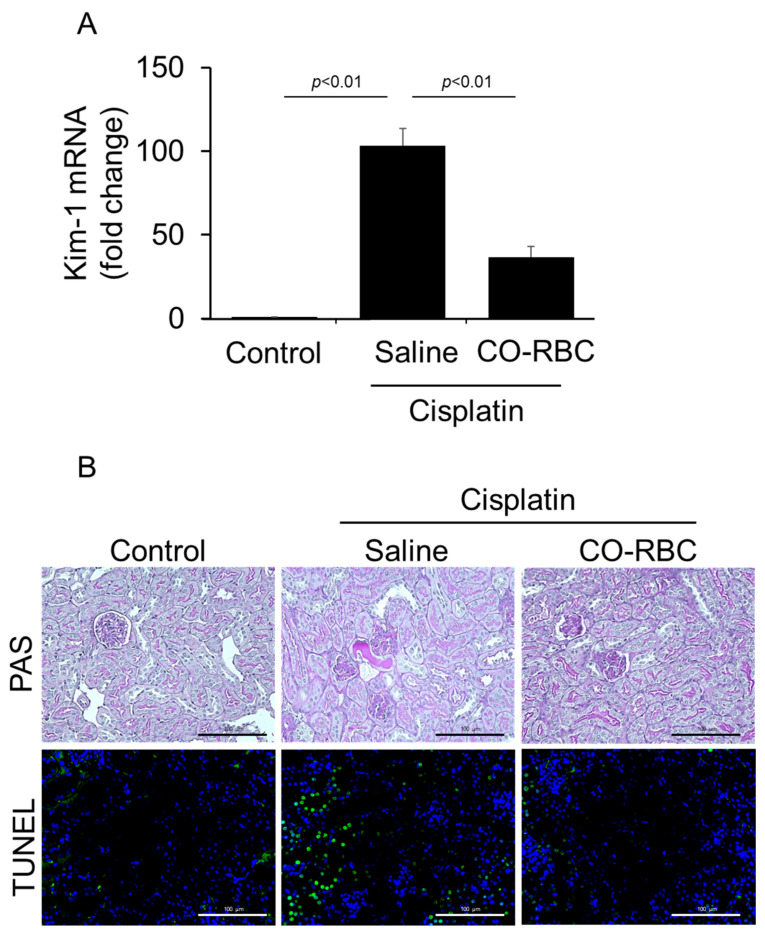
CO-RBC exerts antioxidant effects on cisplatin-induced AKI mice. To examine the effect of CO-RBC on tubular cell injury and histological features, kidneys were collected after treatment with cisplatin combined with saline or CO-RBC. The kidney was analyzed by (**A**) qRT-PCR (*n* = 5/group), (**B**) PAS staining, and TUNEL staining (magnification, ×400, scale bars, 100 µm). Results expressed as the mean ± S.E.M.

**Figure 5 antioxidants-12-01705-f005:**
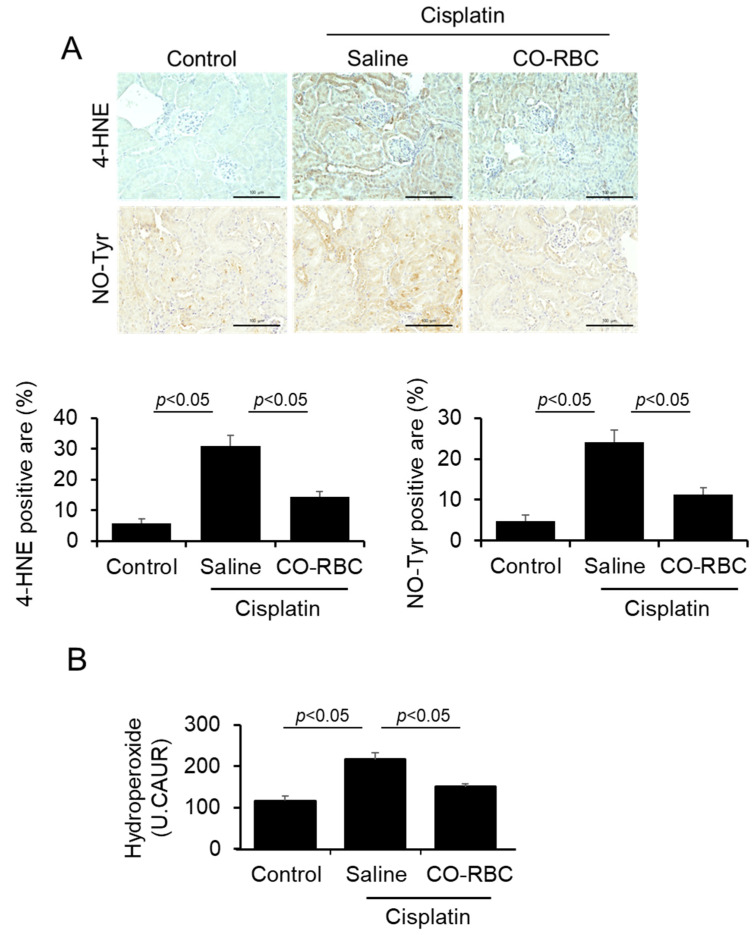
CO-RBC exerts antioxidant effects on cisplatin-induced AKI mice. To assess the antioxidant effects of CO-RBC, kidneys were collected after treatment of cisplatin combined with saline or CO-RBC. The kidney was analyzed by (**A**) immunostaining with antibodies against 4-HNE and NO-Tyr (magnification, ×400, scale bars, 100 µm) and (**B**) d-ROMs test (*n* = 5/group). Results expressed as the mean ± S.E.M.

**Figure 6 antioxidants-12-01705-f006:**
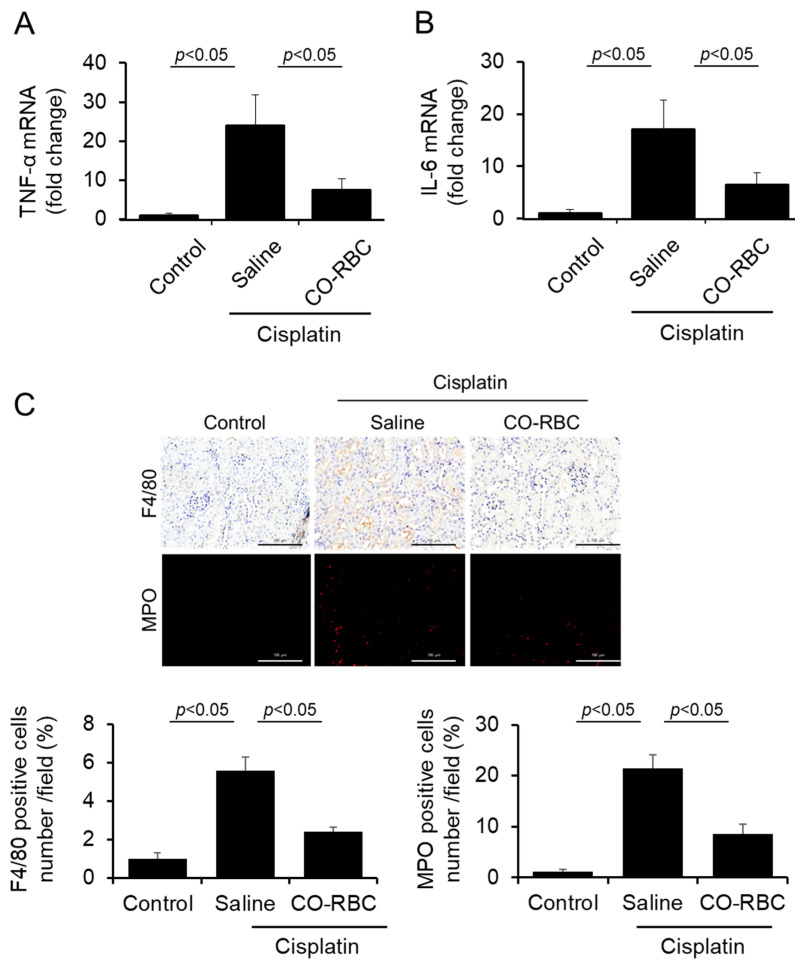
CO-RBC exerts anti-inflammatory effects in cisplatin-induced AKI mice. To investigate the antioxidant effects of CO-RBC, kidneys were collected after treatment with cisplatin combined with saline or CO-RBC. The kidney was analyzed by (**A**,**B**) qRT-PCR (*n* = 5/group) and (**C**) immunostaining with antibodies against F4/80 and MPO (magnification, ×400, scale bars, 100 µm). Results expressed as the mean ± S.E.M.

**Figure 7 antioxidants-12-01705-f007:**
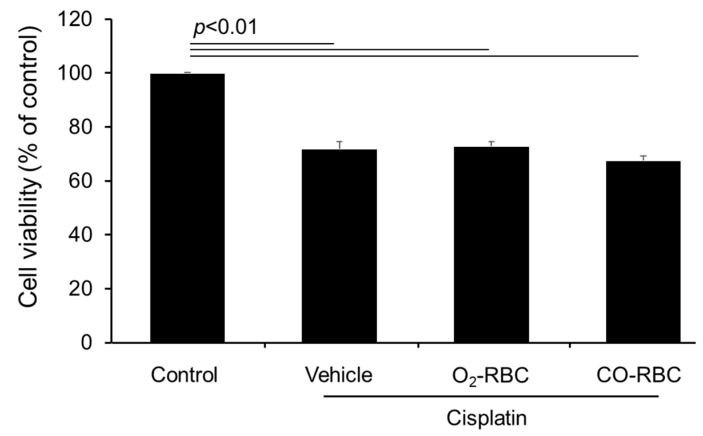
CO-RBC does not weaken the anti-tumor effect of cisplatin. To determine whether CO-RBC affects the anti-tumor effects of cisplatin, B16-F10 melanoma cells were co-treated with 100 μM cisplatin and vehicle or O_2_-RBC (1.0 × 10^7^ cells/mL), CO-RBC (1.0 × 10^7^ cells/mL) for 24 h, then analyzed by a WST-8 assay (*n* = 3/group). Results are expressed as the mean ± S.E.M.

## Data Availability

All of the data is contained within the article and the Appendix A.

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
