# Peer review of "Carbon Monoxide-Loaded Red Blood Cell Prevents the Onset of Cisplatin-Induced Acute Kidney Injury"

_antioxidants, 2023, doi:10.3390/antiox12091705_

Round 1
Reviewer 1 Report (Previous Reviewer 3)
The manuscript was adequately revised based on the previous comments from the reviewer.
The reviewer would like to request a minor change.
-3.7. No effect of CO on the anti-tumor effect was shown in only in vitro conditions. Thus, the authors should mention this limitation in the result and discussion.
Author Response
Please see the attachment.

Reviewer 2 Report (New Reviewer)
Nagasaki et al. used CO-RBC to suppress the impact of ROS on renal disease, and this approach is quite innovative. They also conducted tests at both the HK2 renal tubular cell level and in vivo, confirming the inhibitory effect of CO-RBC on ROS and its protective role in the kidneys. The experimental design by the authors is well-constructed. However, there are still some weaknesses:
1. The rationale for selecting doses of 1.0 √ó 107 cells/ml and 1,400 mgHb/kg for cell and in vivo experiments respectively, is not described by the authors. Please provide detailed explanations for the basis of these dose selections, as well as the correlation between the doses used in cell experiments and those in in vivo experiments. Additionally, clarify if there is a correlation between mouse dosages and human dosages in relation to body weight.
2. I apologize for my limited familiarity with this technique, but from Ogaki et al.'s work (Drug Metab Dispos. 2013), it seems that attention has been given primarily to the dosage of RBC administration, while the amount of CO contained in each RBC has not been detected. Since it's believed that CO is the main active component, it's important to determine the dosage of CO in each experiment for reproducibility. Please correct me if I'm mistaken.
3. The authors mentioned the toxic risks associated with CO in the manuscript. Therefore, I suggest that the authors test the safety of CO-RBC in vivo. Assess whether mice exhibit any signs of CO toxicity, such as cerebral hypoxia, etc. (I noticed in the Materials and Methods section that the authors measured CO levels in the brain, but I didn't see relevant data in the main text.) This is important as if 1,400 mgHb/kg is directly administered to an adult human, a 60kg individual would receive 84gHb, and the contained CO content is not clearly specified.
4. Since CO-RBC mainly alleviates renal damage induced by Cisplatin-generated ROS, it's crucial to measure ROS levels in the kidneys. The authors used the H2DCFDA assay to detect ROS in cell experiments. However, H2DCFDA can also be used for in vivo detection. If the same method is employed for both cell and in vivo experiments, it would be more meaningful to make a comparison. I recommend that the authors use H2DCFDA to measure in vivo ROS levels.
5. For immunohistochemistry or immunofluorescence, please perform statistical analyses. Looking at the authors' figures, such as F4/80, the complete absence of expression in the control group's kidneys is not very convincing. It's possible that the authors inadvertently chose a field of view with no F4/80 signal. Therefore, I suggest that the authors select multiple fields of view to statistically analyze these immunohistochemistry or immunofluorescence data.
Round 2
Reviewer 2 Report (New Reviewer)
The author answered all my doubts with sufficient experimental data and references, so I think the paper has now met the requirements for publication.
This manuscript is a resubmission of an earlier submission. The following is a list of the peer review reports and author responses from that submission.
Round 1
Reviewer 1 Report
Review report on antioxidants-2044597
Manuscript ID: antioxidants-2044597
Type of manuscript: Article
Title: Carbon monoxide-loaded red blood cell prevents the onset of cisplatin-induced acute kidney injury
Authors: Taisei Nagasaki *, Hitoshi Maeda *, Hiroki Yanagisawa, Kento Nishida, Kazuki Kobayashi, Naoki Wada, Isamu Noguchi, Kazuaki Taguchi, Hiromi Sakai, Junji Saruwatari, Hiroshi Watanabe, Masaki Otagiri, Toru Maruyama *
Major Comments:
Cisplatin is a very common chemotherapeutic agent for the treatment of various cancers. After the import through cell membranes via organic cation transporter 2 (OCT2), it is localized in the nucleus and binds to DNA to form an intrastrand DNA complex, which triggers the apoptosis of cancer cells. However, it has a significant drawback for its continuous usage. Prolonged treatment with cisplatin will cause acute kidney injury (AKI), a common clinical problem. Since the main cause of the cisplatin-induced death of cancer cells and the cisplatin-induced AKI is the same, i.e., oxidative stress and inflammation, appropriate means are highly desirable to solve this dilemma. In the present study, Nagasaki et al. reported that carbon monoxide(CO)-loaded red blood cells (CO-RBC) showed a significant protective effect against cisplatin-induced AKI by releasing low concentration of CO in renal by using both human renal proximal tubule epithelial cells (HK-2 cells). CO is known as a silent-killer due to its high affinity for hemoglobin(Hb) in red blood cell, resulting in “CO poisoning”. However, recent studies showed that lower concentrations of CO exerts antioxidant and anti-inflammatory effects. These previous reports are very interesting since CO is proposed as an endogenous signaling molecule produced by heme oxygenase(HO)-mediated heme degradation process with Hb as its major source of supply. Further, Nagasaki et al. found that cisplatin-induced cytotoxicity via oxidative stress and inflammation was suppressed by the CO-RBC treatment by using a mouse model. In the body level of mouse model, administration of CO-RBC significantly suppressed the elevations of blood urea nitrogen and serum creatinine levels caused by intraperitoneal administration of cisplatin. These results suggest significant therapeutic potentials of CO-RBC for the prevention of cisplatin-induced AKI and, further, will improve the usefulness of cisplatin-based chemotherapy.
In recent years, this research group has delt with CO-RBC for the therapy or prevention of acute kidney injury (AKI) or hepatic injury induced by various causes including crush syndrome, rhabdomyoliysis, hemorrhage shock. In each study, they showed promising results for CO-RBC to have a role for providing lower concentrations of CO to kidney cells (or hepatic cells) and to exerts antioxidant and anti-inflammatory effects, although the exact molecular mechanism underlying this ability of CO is still not well understood.
Therefore, present results are very interesting and very important for the expansion towards clinical use of this technique. This report will also help the understanding of molecular mechanism of CO on the antioxidant and anti-inflammatory effects to be solved in near future. Accordingly, this manuscript can be acceptable for publication in “Antioxidant” in the present form.

Reviewer 2 Report
Line 59: please spell out TNF-a and IL-6 before using the abbreviation
Lines 67-80 It would be helpful to have more discussion around any metabolic effects and/or pathways proposed that suggest CO-RBC can help is reducing AKI
Discussion:
The discussion presents findings from the authors’ research study and mentions that CO-RBC provided a renoprotective effect. What could strengthen the discussion and better support the conclusion is how the treatment groups compared to the control group. These comparisons in the discussion can help provide a clearer picture of any possible renoprotective effects.
Reviewer 3 Report
This study examined the renoprotective effect of CO-loaded RBC infusion against cisplatin-induced acute kidney injury. The reviewer raised several issues.
1. The renoprotective mechanism of CO-RBC treatment was not convincing. In in vitro experiment, the protective effect of CO-RBC observed in HK-2 cells is subtle in contrast to the robust renoprotective effect in the mice study. Thus, it is questionable whether the effect of CO-RBC directly targets the damaged tubules. In the pathophysiology of AKI, other cell components including immune cells and endothelium are also involved. If CO-RBS targets other types of cells (not tubules), the in vitro experiment using HK2 would not be adequate. In addition, there is a possibility that the pretreatment of CO-RBC induced some cell protective/antioxidant signaling against cisplatin damage as a precondition manner in a certain type of cell. The authors did not address this point. Although several markers (gene expression and IHC) were evaluated in the animal samples, these show the consequence of kidney damage and do not clarify the protective mechanism of CO-RBC.
2. The authors did not exclude the possibility that CO-RBC infusion modulates the pharmacological kinetics of cisplatin and its metabolite.
3. This study did not examine the CO-RBC effects of the anti-tumor action of cisplatin. If CO-RBC also attenuates the anti-tumor effect of cisplatin, it would not be applied for a therapeutic perspective.